# Development and validation of a depression risk prediction nomogram for US Adults with hypertension, based on NHANES 2007–2018

**Yicheng Wang**[1,2,3], **Yan Zhang** [1,2,3]*, **Binghang Ni**[1,2,3], **Yu Jiang**[1,2,3], **Yu Ouyang**[1,3]

**1** Department of Cardiovascular Medicine, Affiliated Fuzhou First Hospital of Fujian Medical University, Fuzhou, Fujian, China, **2** Fujian Medical University, The Third Clinical Medical College, Fuzhou, Fujian, China, **3** Cardiovascular Disease Research Institute of Fuzhou City, Fuzhou, Fujian, China

* 18558750600@163.com

## Abstract

Depression is of increasing concern as its prevalence increases. Our study's objective was to create and evaluate a nomogram to predict the likelihood that hypertension patients may experience depression. 13293 people with hypertension who were under 20 years old were chosen from the National Health and Nutrition Examination Survey (NHANES) database between 2007 and 2018 for this study. The training and validation sets were split up into the dataset at random in a 7:3 ratio. To find independent predictors, univariate and multivariate logistic regression were employed on the training set. Using information from the validation set, nomogram was subsequently created and internally validated. The effectiveness of the nomogram is assessed using calibration curve and receiver operator characteristic (ROC) curve. Combining univariate logistic regression analysis and multifactor logistic regression analysis, the results showed that age, sex, race, marital, education level, sleep time on workdays, poverty to income ratio, smoking, alcohol consumption, sedentary time and heart failure status were risk factors for hypertensive patients suffering from depression and were included in the nomogram model, and ROC analysis showed that the AUC of the training set was 0.757 (0.797–0.586), with a sensitivity of 0.586; the AUC of the test set was 0.724 (0.712–0.626), with a sensitivity of 0.626, which was a good fit. Decision curve analysis further confirms the value of nomogram for clinical application. In the civilian non-institutionalized population of the United States, our study suggests a nomogram that can aid in predicting the likelihood of depression in hypertension patients and aiding in the selection of the most effective treatments.

## Introduction

Depression is a relatively common mental disease characterized by a persistently negative attitude, decreased interest, and lack of enjoyment [1]. Depression is the leading cause of mental health-related disease burden and the leading cause of disability globally [2]. It affects about 300 million people worldwide [3]. About 8% of American adults are affected by depression [4]. Studies from middle- and high-income nations have indicated that elderly adults are more

**Funding:** Thank you for stating the following in the Funding Section of your manuscript: "This work was supported by the Fuzhou Key Specialty Project (Grant number 20191005),and the Fuzhou "14th Five-Year Plan" Clinical Specialty Training and Cultivation Construction Project, (Grant number 20220103).

**Competing interests:** The authors have declared that no competing interests exist.

likely than teens to experience depression [5]. But among young people, the prevalence of depression has increased significantly from what it was before [6].

Hypertension is one of the most common cardiovascular risk factors [7]. There is evidence that hypertension is strongly associated with depression [8]. Compared to non-hypertension patients, hypertensive patients have a much higher prevalence of unpleasant feelings [9]. Depression is a known independent risk factor for hypertension and is more prevalent in people with hypertension [10]. Patients with other cardiovascular diseases have higher levels of depression than the general population [11]. Sociodemographic characteristics such as gender, race, income, education, age and marital status have been reported to be independently associated with depression [12,13]. Some studies have also found a strong relationship between sleep problems and depression [14]. Chronic conditions such as cancer and diabetes can also cause depression [15,16]. Depression is also associated with smoking, alcohol consumption and other lifestyle habits [17,18].

In cancer and medicine, the nomogram is a helpful tool for determining prognosis. It enables estimations of the likelihood of a single endpoint event to be used in place of the usual prediction model formulas [19]. At present, few clinical predictive models exist for the risk of depression in hypertensive patients. This study used the National Health and Nutrition Examination Survey (NHANES) database to develop and validate a nomogram for predicting the risk of depression in the US adult hypertensive population.

## Methods

### Study design and participants

The National Health and Nutrition Examination Survey (NHANES) is a data set of studies designed to assess the health and nutritional status of the American population. The NHANES employs a complex, multistage probability sample design and conducts surveys of 5000 people every two years on average. By cleaning the data and deleting missing samples, we selected NHANES (2007–2018) as the original dataset for analysis. Each participant in the study provided written informed consent. Due to the open availability of NHANES, no ethical review is required. Six NHANES survey cycles, from 2007 to 2018, were the subject of our analysis.

The ACC/AHA recommends a two-stage classification system for blood pressure, with hypertension being defined as systolic blood pressure (SBP) greater than 130 mmHg and/or diastolic blood pressure (DBP) less than 80 mmHg [20]. Adults with hypertension 20 years of age and older who participated in demographic surveys, physical exams, and questionnaires made up our study population, and the exclusion criteria were: (1) those younger than 20 years of age or younger. (2) those without a diagnosis of hypertension. (3) those with missing values for at least one of all variables. We eliminated 46451 subjects for a variety of reasons, including lack of education, smoking, alcohol use, poverty-to-income ratio (PIR), diabetes, chest pain, coronary heart disease, heart failure, myocardial infarction, stroke, cancer, body mass index (BMI), marital status, ethnicity, sleep time on workdays, waist circumference (WC), sedentary time and age less than 20 and self-reported never diagnosed with hypertension. 13293 participants made up our final study population. The flow chart of the study population screening is shown in **Fig 1**.

### Ethics of approval statement

The authors are accountable for all aspects of the work in ensuring that questions related to the accuracy or integrity of any part of the work are appropriately investigated and resolved. The study was conducted in accordance with the Declaration of Helsinki (as revised in 2013). All

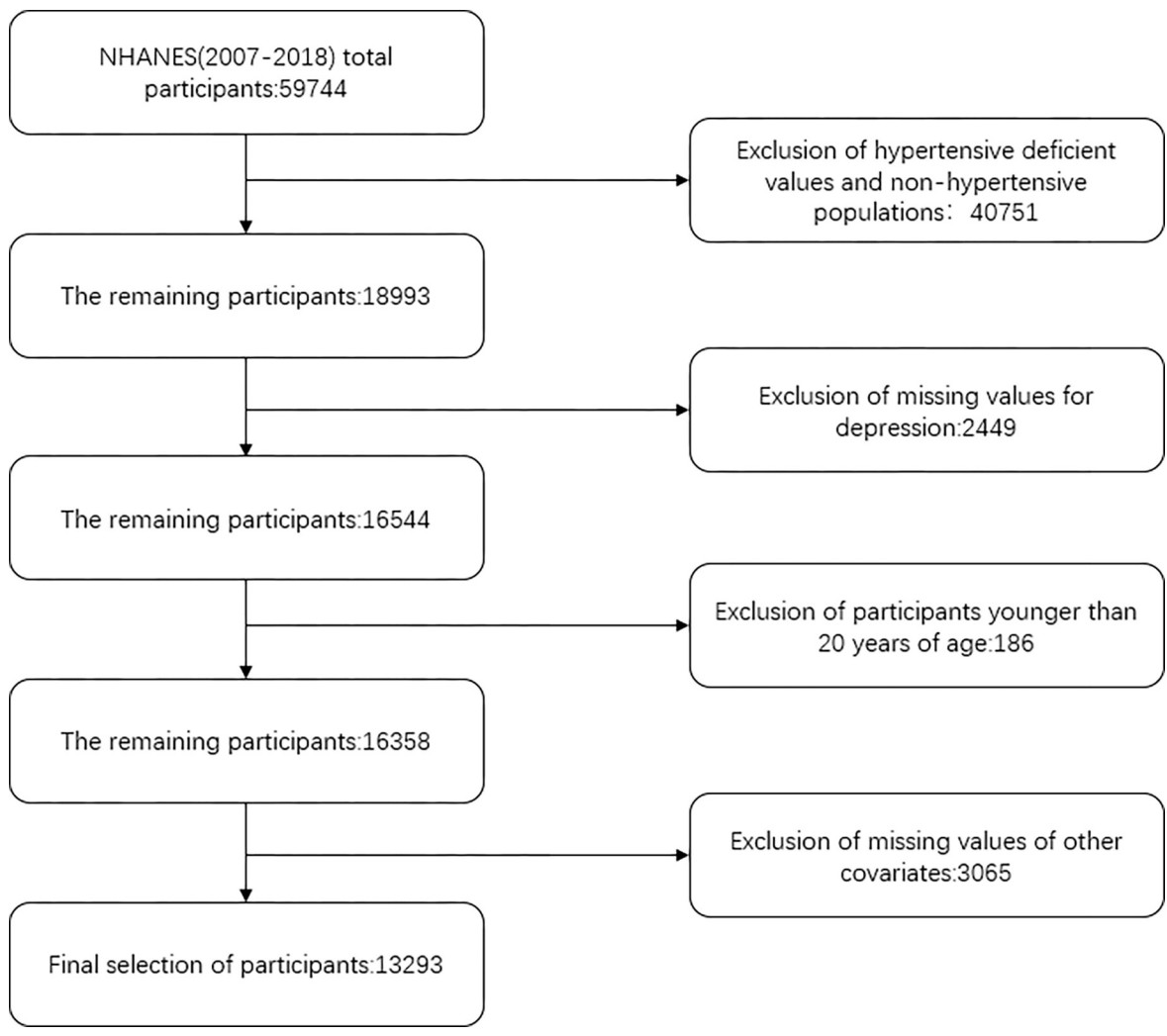

**Fig 1. The flowchart of participants.**

information from the NHANES program is available and free for public, so the agreement of the medical ethics committee board was not necessary.

## Data selection and measurements

We assessed depression using the PHQ-9 scale, a 9-item questionnaire that measures depression on a four-point Likert scale (0 = "not at all", 1 = "a few days", 2 = "more than half the days", or 3 = "almost every day", ranging from 0–27), and depression scores of 10 and above are usually considered to be depressive [21].

Participant demographic characteristics were obtained through a self-reported questionnaire. The demographic characteristics can be categorized by gender (male, female), race (Mexican American, non-Hispanic white, non-Hispanic black, Hispanic, other race), marital status (unmarried, married or living with partner, married but currently living alone [separated, divorced, or widowed]), and education level (below 9th grade, 9th -11th grade, high school graduate, partial college or AA graduate or higher). Participants were asked if they had ever been informed that they had diabetes, coronary heart disease, chest pain, heart failure,

myocardial infarction, or stroke as part of the self-reported medical history. Information about smoking and drinking was also included in the questionnaire part. Each participant's smoking status was assessed by self-report, and they were divided into three groups: nonsmokers, former smokers, and current smokers. Drinking status was categorized according to the degree of drinking as never drinking, formerly drinking now abstaining, heavy drinking (≥3drinks per day for females/≥4 drinks per day for males/binge drinking on 5 or more days per mouth), moderate drinking (≥2drinks per day for females/≥3drinks per day for males/binge drinking ≥2 days per mouth) and light drinking (not including the above). Weight in kilograms divided by height in meters squared (kg/m2) was used to compute BMI. By measuring, the waist circumference is determined. The individuals were questioned, "How much time do you typically sit in a day?" to determine the amount of sedentary time. By dividing household (or individual) income by the survey year and state-specific poverty thresholds, the poverty-to-income ratio was determined. The sleep time on workdays was determined by asking participants, "How much sleep do you typically receive on a workday?".

## Statistical analysis

Continuous variables were expressed using the mean and Standard error. For categorical variables that were expressed as frequencies or percentages, chi-square tests or Fisher exact tests were used to determine group differences. In hypertensive patients, we used univariate and multivariate logistic regression to identify risk factors for the emergence of depression. Due to the complex sampling characteristics of the NHANES, we used NHANES sample weights in logistic regression analyses to obtain nationally representative estimates. As effect estimates, the ratio (OR) and 95% confidence interval (CI) were used.

A random 7:3 split was made between the training and validation sets. The training set was utilized to identify independent predictors of depression using univariate and multivariate logistic regression. We employed the ratio to evaluate the relationship between the output of the outcome model and the predictors. The internal validation of a nomogram was then performed using the data from the validation set. For the prediction models created from the training and validation sets, the area under the receiver operating characteristic curve (AUC) values were computed to assess the stability of the models. Utilizing receiver operating characteristic curves, the nomogram's performance was assessed.

To evaluate the calibration capabilities, a calibration graph, which is a graph with specific parameters, is employed. The ideal line on the calibration graph is at a 45 degree angle, and the actual and anticipated probabilities are equal. If the equations for the graph are drawn along the ideal line, the generated nomogram is thought to have accurate prediction capability. In order to identify the greatest benefit of the prediction model, we use the decision curve analysis (DCA) method. $P < 0.05$ was regarded as significant for all analyses, which were conducted using R software (Version 4.2.2).

## Results

### Baseline clinical characteristics

The clinical characteristics of the weighted study population are shown in **Table 1**. In the training and validation groups, there were 902 and 367 patients with depression out of a total of 13,293 patients with hypertension, respectively. Of the 1269 patients with depression, 493 were male and 776 were female. The mean age of the entire population was 54.06 years, the mean BMI was 30.80 kg/m2, the mean poverty to income ratio was 3.04, the mean sleep time on workdays was 7.11 hours, the mean sedentary time was 378.38 minutes, and the mean waist circumference was 104.82cm. Except for race and cancer, there were statistically significant

**Table 1. Baseline characteristics in training and validation cohorts.**

| Variables | Total | Training Depression | set Non-depression | P-value | Testing Depression | set Non-depression | P-value |
|---|---|---|---|---|---|---|---|
| N | 13293 | 902 | 8404 | | 367 | 1252 | |
| AGE (years) | 54.08(0.25) | 52.06(0.64) | 54.24(0.27) | 0.002 | 51.46(0.93) | 54.39(0.39) | 0.004 |
| BMI (kg/m$^2$) | 30.80(0.09) | 32.21(0.35) | 30.68(0.11) | <0.001 | 32.64(0.54) | 30.58(0.15) | <0.001 |
| PIR | 3.04(0.04) | 2.10(0.10) | 3.13(0.04) | <0.001 | 2.17(0.13) | 3.13(0.04) | <0.001 |
| SLEEP (hours) | 7.11(0.02) | 6.67(0.09) | 7.15(0.02) | <0.001 | 6.78(0.13) | 7.14(0.03) | 0.01 |
| Sedentary (minutes) | 378.38(3.01) | 404.86(10.05) | 375.98(3.42) | 0.01 | 386.89(15.98) | 377.28(4.53) | 0.55 |
| WC (cm) | 104.82(0.21) | 107.30(0.75) | 104.67(0.27) | <0.002 | 107.94(1.27) | 104.31(0.34) | 0.01 |
| Education | | | | <0.001 | | | 0.002 |
| Less than 9$^{th}$ grade | 1369 (10.3%) | 131(8.11%) | 801(4.59%) | | 55(8.10%) | 382(4.96%) | |
| 9-11$^{th}$ grade | 1916 (14.41%) | 180(16.22%) | 1143(10.11%) | | 84(18.42%) | 509(10.30%) | |
| High school graduate | 3233 (24.32%) | 228(27.38%) | 2059(24.09%) | | 75(26.99%) | 871(25.48%) | |
| Some college graduate | 3969 (29.86%) | 268(35.85%) | 2534(32.95%) | | 108(27.84%) | 1059(31.94%) | |
| College graduate Or above | 2806 (21.11%) | 95(12.43%) | 1867(28.26%) | | 45(18.65%) | 799(27.32%) | |
| MARITAL | | | | <0.001 | | | <0.001 |
| Never married | 1657(12.47%) | 141(15.67%) | 996(11.79%) | | 69(18.90%) | 451(12.50%) | |
| Living with Partner | 7955(59.84%) | 384(46.26%) | 5213(66.43%) | | 159(51.49%) | 2199(66.21%) | |
| Widowed/Divorced | 3681(27.69%) | 377 (38.07%) | 2195(21.78%) | | 139(29.62%) | 970(21.29%) | |
| RACE | | | | 0.05 | | | 0.04 |
| Non-Hispanic White | 5816(43.75%) | 414(67.87%) | 3661(70.52%) | | 150(61.16%) | 1591(70.41%) | |
| Non-Hispanic Black | 3258(24.51%) | 198(13.05%) | 2079(12.01%) | | 96(15.67%) | 885(12.35%) | |
| Mexican American | 1699(12.78%) | 119(6.74%) | 1056(6.50%) | | 52(6.59%) | 472(6.42%) | |
| Other Hispanic | 1226(9.22%) | 114(6.53%) | 757(4.46%) | | 41(5.61%) | 314(4.15%) | |
| Other race | 1294(9.73%) | 57(5.81%) | 851(6.51%) | | 28(10.97%) | 358(6.66%) | |
| SEX | | | | <0.001 | | | <0.001 |
| Female | 6263(47.12%) | 555(61.24%) | 3798(45.26%) | | 221(61.38%) | 1689(46.63%) | |
| Male | 7030(52.88%) | 347(38.76%) | 4606(54.74%) | | 146(38.62%) | 1931(53.37%) | |
| DM | | | | 0.015 | | | 0.04 |
| No | 8324(62.62%) | 501(61.20%) | 5312(67.78%) | | 217(66.72%) | 2294(67.89%) | |
| Prediabetes | 1339(10.07%) | 77(9.45%) | 847(10.44%) | | 27(7.23%) | 388(11.14%) | |
| Yes | 3630(27.31%) | 324(29.35%) | 2245(21.77%) | | 123(26.06%) | 938(20.97%) | |
| SMOKE | | | | <0.001 | | | <0.001 |
| Never | 6836(51.43%) | 353(38.48%) | 4416(52.74%) | | 133(33.47%) | 1934(52.84%) | |
| Former | 3849(28.96%) | 230(25.01%) | 2455(29.95%) | | 96(28.03%) | 1068(31.68%) | |
| Now | 2608(19.62%) | 319(36.52%) | 1533(17.31%) | | 138(38.50%) | 618(15.48%) | |
| ALCOHOL | | | | <0.001 | | | 0.12 |
| Never | 1893(14.24%) | 109(9.24%) | 1206(10.51%) | | 44(7.45%) | 534(10.79%) | |
| Former | 2473(18.6%) | 220(23.36%) | 1504(14.49%) | | 84(20.27%) | 534(10.79%) | |
| Mild | 4743(35.68%) | 242(29.88%) | 3077(40.41%) | | 111(33.00%) | 1313(39.30%) | |
| Moderate | 1842(13.86%) | 118(14.58%) | 1185(16.07%) | | 52(16.18%) | 487(15.87%) | |
| Heavy | 2342(17.62%) | 213(22.95%) | 1432(18.53%) | | 76(23.10%) | 621(19.05%) | |
| STROKE | | | | <0.001 | | | <0.001 |
| No | 12583(94.66%) | 809(91.88%) | 7994(96.12%) | | 331(91.82%) | 3449(96.44%) | |

*(Continued)*

**Table 1.** (Continued)

| Variables | Total | Training Depression | set Non-depression | P-value | Testing Depression | set Non-depression | P-value |
|---|---|---|---|---|---|---|---|
| Yes | 710(5.34%) | 93(8.12%) | 410(3.88%) | | 36(8.18%) | 171(3.56%) | |
| CANCER | | | | 0.77 | | | 0.60 |
| No | 11595(87.23%) | 775(86.60%) | 7339(86.13%) | | 322(88.19%) | 3159(86.76%) | |
| Yes | 1698(12.77%) | 127(13.40%) | 1065(13.87%) | | 45(11.81%) | 461(13.24%) | |
| MI | | | | <0.001 | | | 0.25 |
| No | 12511(94.12%) | 802 (91.42%) | 7934(95.35%) | | 339(94.43%) | 3436(95.79%) | |
| Yes | 782(5.88%) | 100(8.58%) | 470(4.65%) | | 28(5.57%) | 184(4.21%) | |
| HF | | | | <0.001 | | | <0.001 |
| No | 12684(95.42%) | 815(91.62%) | 8052(96.69%) | | 332(93.23%) | 3485(97.03%) | |
| Yes | 609(4.58%) | 87(8.38%) | 352(3.31%) | | 35(6.77%) | 135(2.97%) | |
| CHD | | | | 0.02 | | | 0.19 |
| No | 12507(94.09%) | 824(92.38%) | 7919(94.82%) | | 335(93.95%) | 3429(95.41%) | |
| Yes | 786(5.91%) | 78(7.62%) | 485(5.18%) | | 32(6.05%) | 191(4.59%) | |
| CHEST PAIN | | | | | | | |
| No | 12809(96.36%) | 836(93.75%) | 8130(97.04%) | <0.001 | 340(95.19%) | 3503(97.05%) | 0.03 |
| Yes | 484(3.64%) | 66(6.25%) | 274(2.96%) | | 27(4.81%) | 117(2.95%) | |

poverty-to-income ratio (PIR); waist circumference (WC); body mass index (BMI); Heart failure (HF); myocardial infarction (MI); coronary heart disease (CHD); diabetes mellitus (DM).

differences between the depressed and non-depressed patients in the training group for age, gender, education, marital status, PIR, BMI, waist circumference, diabetes mellitus status, coronary heart disease status, myocardial infarction status, chest pain status, heart failure status, stroke status, sleep time on workdays, smoking and alcohol consumption status (P<0.05). There were statistically significant differences between depressed patients in the validation group and non-depressed patients in terms of age, gender, race, education, marital status, PIR, BMI, waist circumference, diabetes status, chest pain status, heart failure status, stroke status, sleep time on workdays, and smoking status(P<0.05).

## Univariate and Multivariate logistic regression

The results of univariate and multivariate logistic regressions are shown in Table 2. Age, gender, BMI, race, education, marital status, poverty-to-income ratio, sleep time on workdays, waist circumference, sedentary time, diabetes status, myocardial infarction status, coronary heart disease status, chest pain status, heart failure status, stroke status, cancer status, smoking, and alcohol consumption were analyzed using univariate logistic regression models for the included population. The results of the study showed that all risk variables for depression were statistically significant (p<0.05), except for cancer and coronary heart disease. Further multivariable logistic regression analysis showed that higher education was associated with a lower risk of depression. Married people (OR: 0.67, 95% CI: 0.48–0.93, p = 0.02) had a significantly lower risk of depression than unmarried and divorced people. Among the various racial groups in the United States, African Americans (OR: 0.60, 95% CI: 0.48–0.75, P<0.001) had the lowest risk of depression. Patients with hypertension (OR: 1.87, 95% CI: 1.25–2.80, P = 0.003) who had previously consumed alcohol and now abstained had the lowest risk of depression. Patients with hypertension who were currently smoking (OR: 2.31, 95% CI: 1.72–3.10, p<0.001) had the highest risk of depression. However, we also found that men (OR: 0.48,

**Table 2. Weighted univariate and multivariate logistic regression analysis of the training set.**

| Variable | Univariate OR (95%CI) | P-value | Multivariate OR (95%CI) | P-value |
|---|---|---|---|---|
| EDU | | | | |
| Less than 9th grade | 1.00 | | 1.00 | |
| 9-11th grade | 0.85(0.61,1.17) | 0.31 | 0.75(0.54,1.04) | 0.08 |
| High school graduate | 0.30(0.20,0.45) | <0.001 | 0.70(0.50,0.99) | 0.04 |
| Some college graduate | 0.65(0.48,0.87) | 0.004 | 0.67(0.49,0.92) | 0.01 |
| College graduate or above | 0.58(0.44,0.76) | <0.001 | 0.58(0.36,0.92) | 0.02 |
| MARITAL | | | | |
| Never married | 1.00 | | 1.00 | |
| Married/Living with Partner | 1.92(1.46,2.53) | <0.001 | 0.67(0.48,0.93) | 0.02 |
| Widowed/Divorced | 2.40(1.90,3.02) | <0.001 | 1.12(0.78,1.61) | 0.50 |
| RACE | | | | |
| Non-Hispanic White | 1.00 | | 1.00 | |
| Non-Hispanic Black | 1.32(1.03,1.70) | 0.03 | 0.60(0.48,0.75) | <0.001 |
| Mexican American | 1.13(0.84,1.51) | 0.42 | 0.89(0.66,1.21) | 0.50 |
| Other Hispanic | 1.27(0.84,1.92) | 0.26 | 1.12(0.84,1.49) | 0.40 |
| Other race | 0.96(0.77,1.18) | 0.68 | 1.38(0.86,2.7) | 0.20 |
| SEX | | | | |
| Female | 1.00 | | 1.00 | |
| Male | 0.52(0.42,0.63) | <0.001 | 0.48(0.39,0.61) | <0.001 |
| AGE | 0.99(0.99,0.99) | <0.001 | 0.99(0.98,1.00) | 0.007 |
| DM | | | | |
| No | 1.00 | | 1.00 | |
| Prediabetes | 0.80(0.56,1.15) | 0.23 | 0.81(0.55,1.21) | 0.30 |
| Yes | 1.43(1.15,1.77) | 0.001 | 1.27(1.00,1.63) | 0.05 |
| SMOKE | | | | |
| Never | 1.00 | | 1.00 | |
| Former | 0.80(0.63,1.01) | 0.06 | 1.27(0.98,1.63) | 0.070 |
| Now | 2.46(1.90,3.20) | <0.001 | 2.31(1.72,3.10) | <0.001 |
| ALCOHOL | | | | |
| Never | 1.00 | | 1.00 | |
| Former | 2.13(1.50,3.04) | <0.001 | 1.87(1.25,2.80) | 0.003 |
| Mild | 1.03(0.77,1.38) | 0.84 | 1.47(1.03,2.10) | 0.03 |
| Moderate | 1.19(0.81,1.74) | 0.38 | 1.27(0.79,2.03) | 0.31 |
| Heavy | 1.61(1.17,2.22) | 0.004 | 1.37(0.87,2.16) | 0.18 |
| STROKE | | | | |
| No | 1.00 | | 1.00 | |
| Yes | 2.00(1.39,2.89) | <0.001 | 1.32(0.87,2.00) | 0.20 |
| HF | | | | |
| No | 1.00 | | 1.00 | |
| Yes | 2.38(1.67,3.39) | <0.001 | 1.58(1.02,2.44) | 0.04 |
| CANCER | | | | |
| No | 1.00 | | 1.00 | |
| Yes | 0.96(0.71,1.30) | 0.81 | 1.11(0.78,1.58) | 0.57 |
| MI | | | | |
| No | 1.00 | | 1.00 | |
| Yes | 1.65(1.21,2.24) | 0.002 | 1.05(0.69,1.61) | 0.80 |
| CHD | | | | |

(*Continued*)

**Table 2.** (Continued)

| Variable | Univariate OR (95%CI) | P-value | Multivariate OR (95%CI) | P-value |
|---|---|---|---|---|
| No | 1.00 | | 1.00 | |
| Yes<br>Chest pain<br>No<br>Yes | 1.18(0.83,1.68)<br>1.00<br>1.88(1.31,2.71) | 0.34<br><0.001 | 0.85(0.54,1.34)<br>1.00<br>1.25(0.75,2.09) | 0.48<br>0.38 |
| BMI | 1.03(1.02,1.05) | <0.001 | 1.01(0.98,1.05) | 0.53 |
| PIR | 0.69(0.63,0.74) | <0.001 | 0.78(0.71,0.85) | <0.001 |
| SLEEP | 0.81(0.74,0.89) | <0.001 | 0.84(0.77,0.91) | <0.001 |
| WC | 1.01(1.02,1.05) | <0.001 | 1.00(0.99,1.02) | 0.48 |
| SEDENTARY | 1.00(1.00,1.00) | 0.01 | 1.00(1.00,1.00) | 0.002 |

poverty-to-income ratio (PIR); waist circumference (WC); body mass index (BMI); Heart failure (HF); myocardial infarction (MI); coronary heart disease (CHD); diabetes mellitus (DM).

95% CI: 0.39–0.61, p<0.001) were less likely to suffer from depression than women. The worse the economic conditions, the higher the risk of depression. The less sleep people get on workdays with hypertension, the greater the risk of depression.

## Development and validation of the nomogram

Based on the results of univariate regression analysis and multivariate regression analysis, predictors such as gender, BMI, race, education level, marital status, PIR, sleep time on workdays, diabetes status, smoking and alcohol consumption were identified, and accordingly, the nomogram model (**Fig 2**) was developed to predict the risk of depression in hypertensive people.

ROC curves for training set and test set were used to evaluate the discriminability of the model. the results of ROC analysis showed that the AUC of the training set was 0.757 (0.797–0.586) with a sensitivity of 0.586, and the AUC of the validating set was 0.724 (0.712–0.626) with a sensitivity of 0.626, indicating that our model has good stability and prediction accuracy. The results are displayed in **Fig 3A** and **3B**.

The calibration plot for training set and test set shows that the predicted and observed probabilities are similar. These results suggest that the nomogram can accurately predict the risk of depression in the study population. The results are presented in **Fig 4A** and **4B**.

As shown in **Fig 5**, the net benefit of the nomogram decision curve is also higher when the threshold value is higher, which can therefore indicate the good clinical usefulness of our nomogram.

## Discussion

In this population-based investigation, we developed and validated a nomogram model to gauge the risk that individuals with hypertension may experience depression. Eleven nomogram predictors were found using logistic regression models, including age, sleep time on workdays, poverty-to-income ratio, education level, marital status, race, sex, heart failure condition, smoking, and alcohol use. The ROC analysis of the training set showed an AUC of 0.757. After internal validation, the AUC of the test set was 0.724, suggesting that our model had an excellent track record of making predictions. The calibration curves accurately depict the calibration results from both the development and validation groups. The decision curve analysis showed that the nomogram has good clinical value.

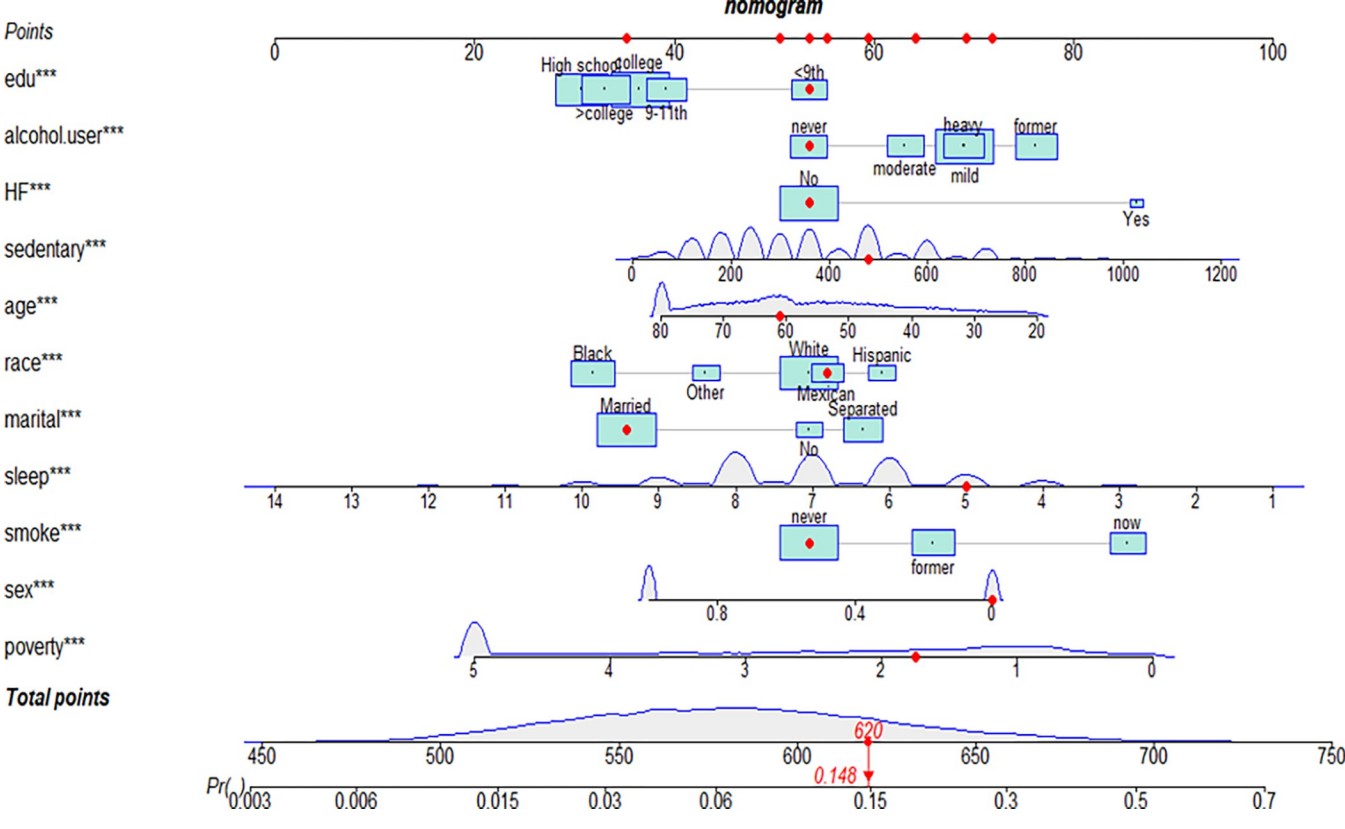

**Fig 2. Nomogram for predicting depression for patients with hypertension.**

According to our study, depression was negatively correlated with the poverty-to-income ratio, indicating that the more financially precarious the family, the greater the chance of getting depression. The effect of higher or lower income levels on depression has been demonstrated in many studies, with low-income groups tending to be more likely to suffer from depression [22,23]. The possible reason is that lower income affects psychosocial factors in low-income people, which increases the risk of depressive symptoms in low-income people [24]. Consequently, policy makers should give more consideration to mental health screening of low-income populations. In our study, blacks were less likely to suffer from depression than whites. The possible reason for this current difference between races is due to differences in the way cultures express themselves between races [25]. Our study concluded that educational attainment was strongly associated with the risk of depression in patients with hypertension, and that the higher the educational attainment, the lower the risk of depression. Similarly, the results of an Indonesian study were similar to our findings [26]. It is currently believed that from a psychological perspective, individuals with higher education are perceived to have better coping and problem-solving skills, which effectively prevent adverse health outcomes [27]. Of course, we also discovered a significant gender difference between men and women with hypertension and their risk of depression. Women with hypertension have an approximately 50% higher risk of developing depression than men with hypertension. Previous research has shown that the gender difference that women are more likely to suffer from depression than men is caused by the secretion of testosterone, which has anxiolytic and antidepressant effects, and that men produce more testosterone and are therefore less likely to suffer from depression than women [28]. The average age of the hypertension participants in our study who also had

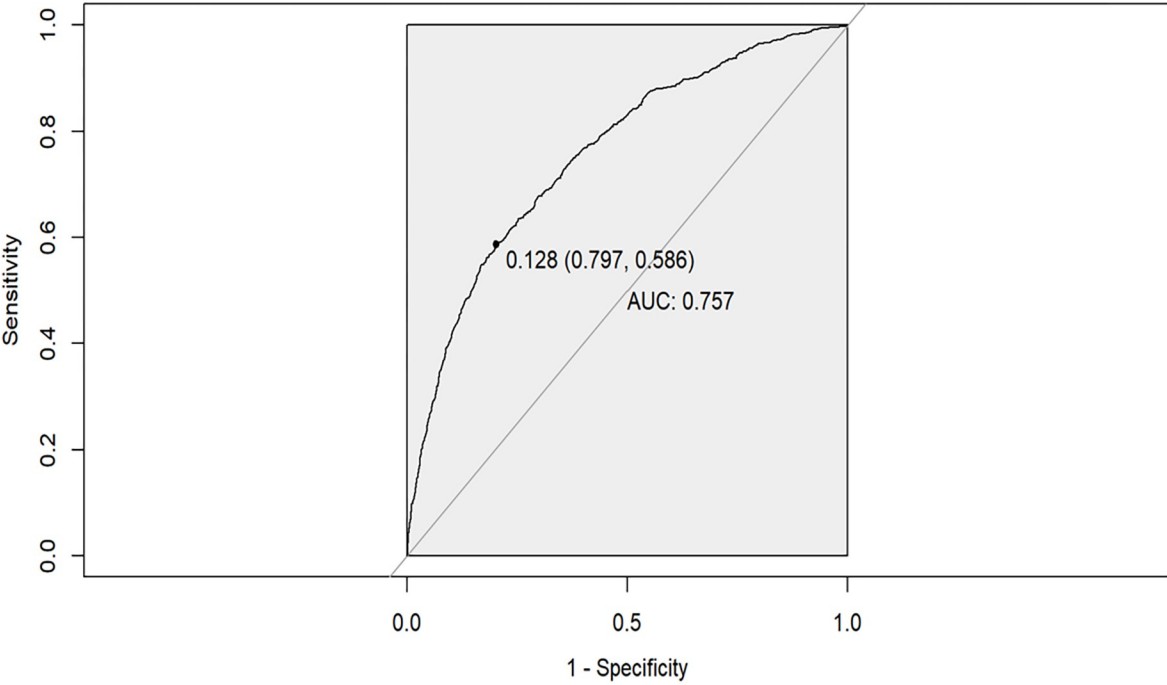

A.

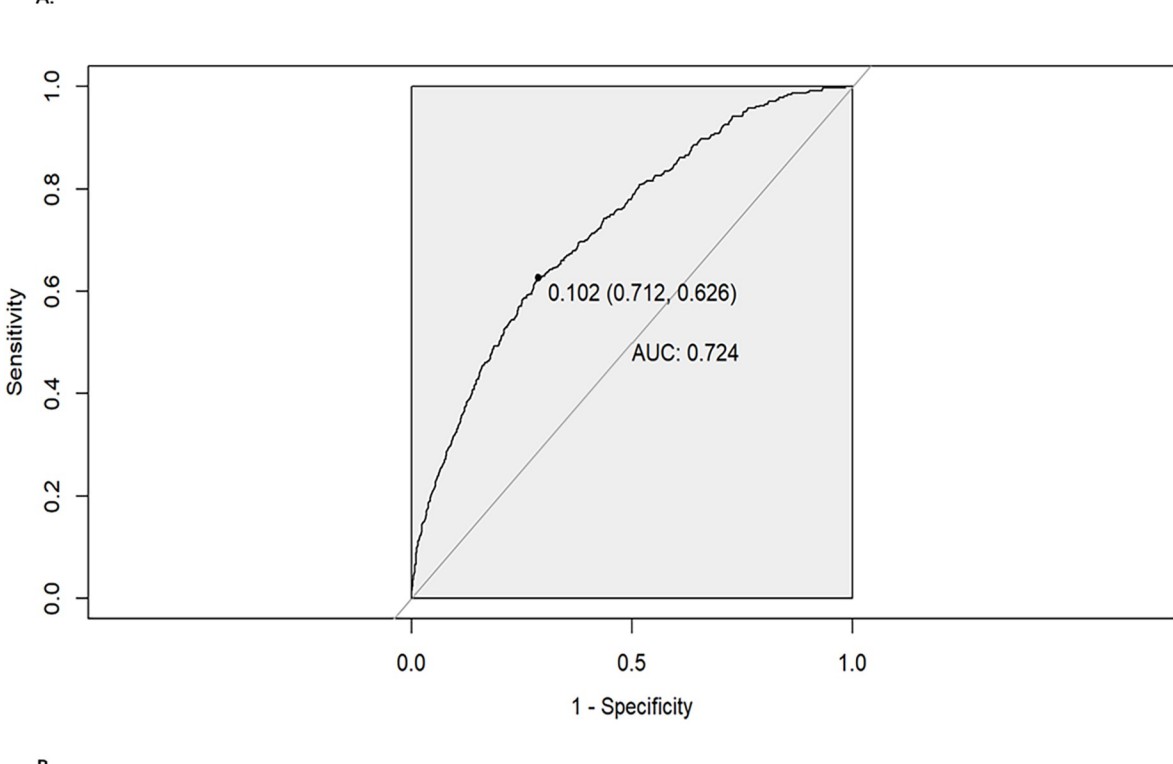

B.

**Fig 3.** A. ROC curve for training set. B. ROC curve for test set.

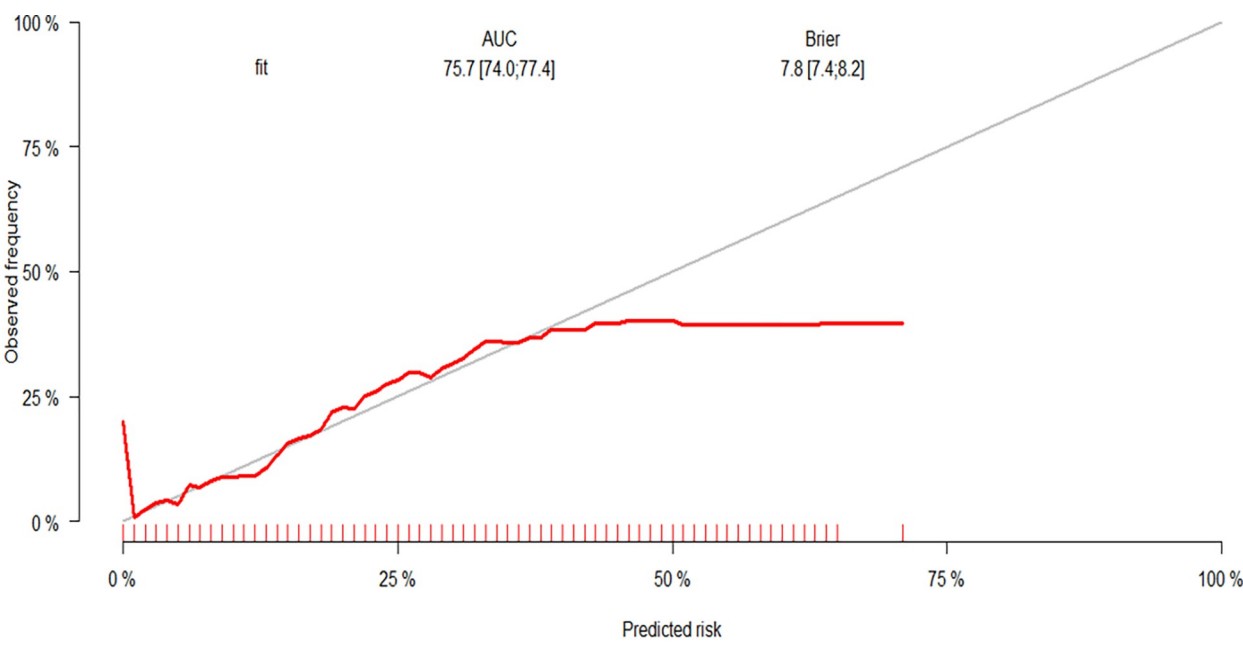

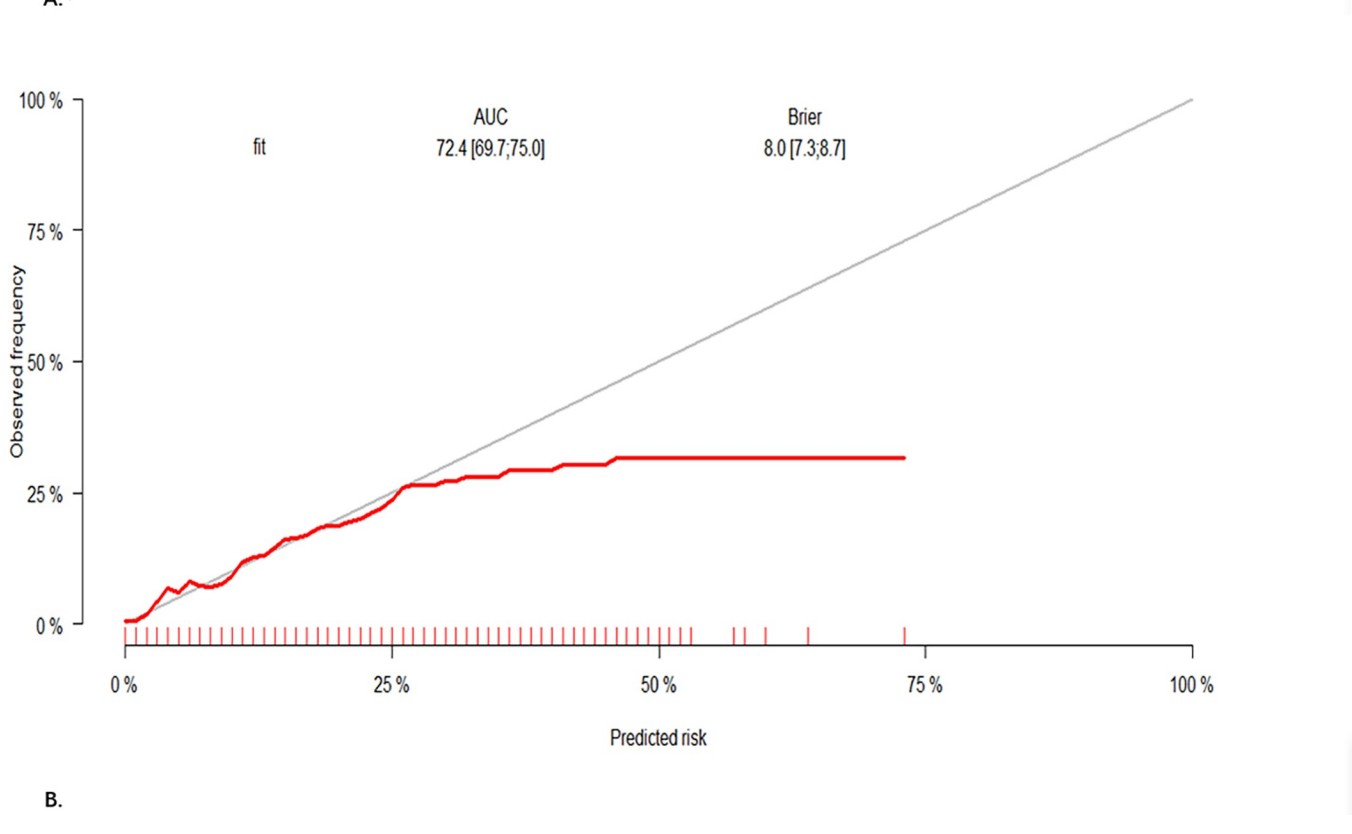

**Fig 4.** A. Calibration curve for training set. B. Calibratio curve for test set.

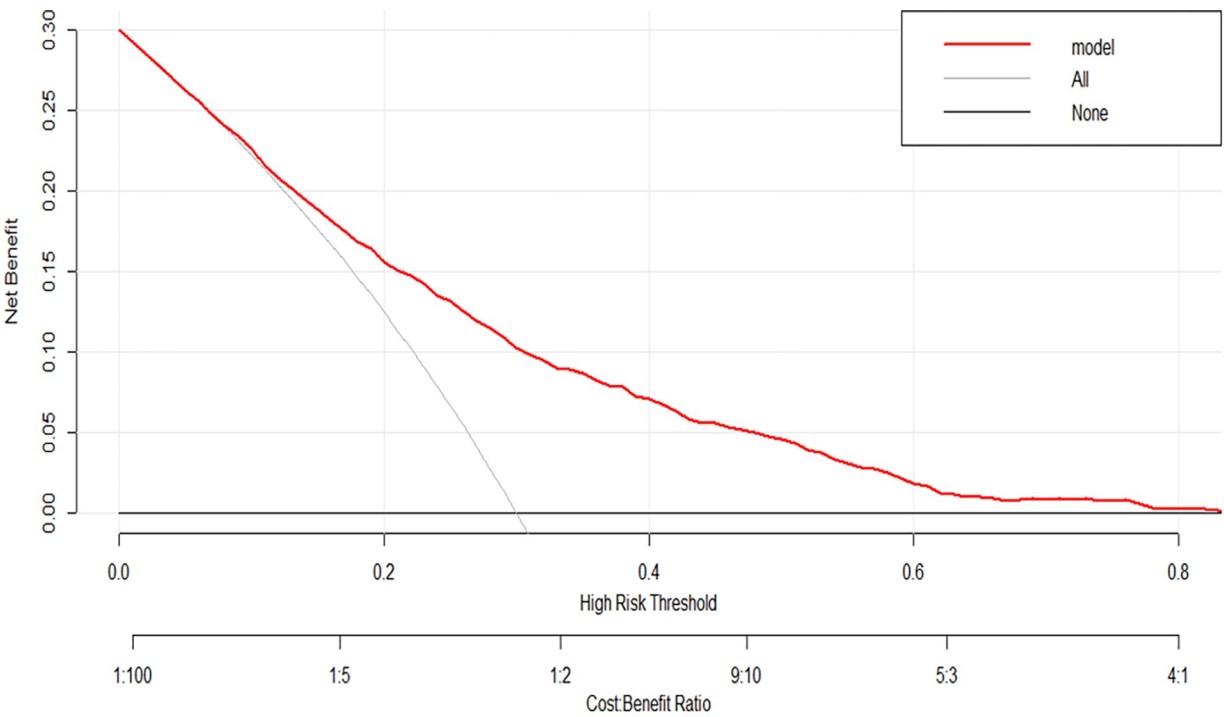

**Fig 5. DCA curve for nomogram.**

depression was 54.08 years old. The hypertensive population that develops depression is more concentrated in middle-aged and older adults. Studies have shown that the older a person is, the more likely they are to develop depression [29,30]. As people age, the cumulative effects of a number of adverse factors may increase the risk of negative mental health outcomes, ultimately leading to depression [31,32]. In subsequent studies, further stratification of different age groups could be done to determine the risk of having depression in different age groups. Combining gender and age differences in susceptibility to depression, it can be hypothesized that middle-aged and older menopausal or postmenopausal women would be at higher risk for depression than other age groups. Also, our research indicates that these non-depressed individuals had smaller BMI and waist circumference than depressed individuals. Such findings have been reported in many studies [33,34]. Among them, the inflammatory response is crucial in the development of depression brought on by obesity [35,36].

A number of other risk factors for depression have also been studied in earlier research. Sleep disorders is an independent predictor of depression [37–39]. Reduced sleep in teenagers raises the chance of major depression, which in turn raises the risk of reduced sleep, according to an analysis of the relationship between sleep deprivation and the risk of depression in adolescents [40]. Likewise, in our study, for persons with hypertension, the shorter their sleep during the workday, the higher their risk of depression. Further research is needed to explore the causal relationship between sleep duration and depression. Additionally, the likelihood of depression in hypertension individuals is significantly influenced by smoking and alcohol use. Many studies have confirmed that smoking is a risk factor for depression [41,42]. According to multivariable logistic regression analysis, smoking increased the prevalence of depression in hypertensive patients, whereas those who used to smoke but now quit smoking had a significantly lower risk of depression compared with hypertensive people who now smoke. Similar to

our results, a meta-analysis showed that quitting smoking reduced the incidence of depression and improved positive mood and quality of life compared to continued smoking [42]. In the present study, drinking status as one of the risk factors for depression was also confirmed in many studies [43–45]. Patients who drink alcohol were more likely to suffer from depression compared to those who did not drink alcohol. In addition, we also observed that patients who lightly drink alcohol were at lower risk of depression compared to abstinent drinkers. A Chinese Mendelian randomization study found that alcohol use was associated with lower levels of depression [46]. There are similar findings in some other studies [47,48]. The mechanisms underlying the beneficial association of alcohol use level and depression are still under debate and further studies are needed to confirm the association. In our analysis, people with a diagnosis of heart failure had a 58% higher risk of depression than people without one. Heart failure is the end-stage manifestation of many cardiovascular diseases [49]. The findings of a number of recent studies confirm the idea that depression is linked to the onset and progression of heart failure [50–52]. We advise routine psychological examinations for the group of chronically sick people in light of this.

Predictor identification and risk assessment are essential for effective medical decision making to prevent depression in patients with hypertension. In patients with hypertension, multifactorial interventions have been shown to have significant benefits in circumventing the risk of depression. Nomogram is a total score based on the values of multiple predictor variables for a person. Clinicians can calculate the risk of an event occurring based on the total score of the nomogram [53]. In developing a model to predict depression in hypertensive populations, we chose variables that are clinically simple and easy to collect, which could improve the clinical utility of the nomogram in this study. Our nomogram is able to obtain the probability of risk of developing depression in hypertensive patients to help clinicians make more favorable treatment decisions for their patients.

However, there are some limitations on our study. First of all, we must note that all of our data came from NHANES health checks at home interviews and ambulatory screening centers, with the exception of some variables for which self-reports were used. It might put some restrictions on how accurate our data is and compromise the objectivity of the results. Secondly, Given that NHANES is a large database with numerous indicators, we were unable to include all covariates associated with hypertension and depression because of our current knowledge inadequacies. As a result, some variables may have been overlooked and subject to selection bias in our selection of included variables. Thirdly, our study lacks external validation, therefore, external datasets are needed to further validate the reliability of the findings. Last, to ascertain how the prediction model performs in various cultural contexts, our study only includes data from the population of the United States and needs to be validated using a large sample from various nations.

## Conclusion

In conclusion, we developed a predictive model to predict the risk of depression in hypertensive patients using data from the NHANES database. The model showed equally good discrimination in different datasets. Nomogram help to screen hypertensive patients for the risk of depression and can be effective in making good clinical decisions for clinicians.

## Supporting information

**S1 File.**
(XLSX)

## Author Contributions

**Data curation:** Yicheng Wang.

**Formal analysis:** Yicheng Wang.

**Software:** Yicheng Wang.

**Writing – original draft:** Yicheng Wang, Yan Zhang.

**Writing – review & editing:** Yan Zhang, Binghang Ni, Yu Jiang, Yu Ouyang.

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
