## [Decision Letter · Decision Letter 0]

7 Feb 2023

PONE-D-22-34347Development and validation of a depression risk prediction nomogram for US Adults with hypertension, based on NHANES 2007-2018PLOS ONE

Dear Dr. Zhang,

Thank you for submitting your manuscript to PLOS ONE. After careful consideration, we feel that it has merit but does not fully meet PLOS ONE’s publication criteria as it currently stands. Therefore, we invite you to submit a revised version of the manuscript that addresses the points raised during the review process. Below you will find the comments from the reviewers which should be addressed.

We look forward to receiving your revised manuscript.

Kind regards,

Gregor Stiglic, Ph.D.

Academic Editor

PLOS ONE

Journal Requirements:

"This work was supported by the Fuzhou Key Specialty Project (Grant number 20191005),and

the Fuzhou "14th Five-Year Plan" Clinical Specialty Training and Cultivation Construction

Project, (Grant number 20220103)."

5. PLOS requires an ORCID iD for the corresponding author in Editorial Manager on papers submitted after December 6th, 2016. Please ensure that you have an ORCID iD and that it is validated in Editorial Manager. To do this, go to ‘Update my Information’ (in the upper left-hand corner of the main menu), and click on the Fetch/Validate link next to the ORCID field. This will take you to the ORCID site and allow you to create a new iD or authenticate a pre-existing iD in Editorial Manager. Please see the following video for instructions on linking an ORCID iD to your Editorial Manager account: https://www.youtube.com/watch?v=_xcclfuvtxQ.

7. Please include a separate caption for each figure in your manuscript.

Reviewers' comments:

Reviewer's Responses to Questions

**Comments to the Author**

1. Is the manuscript technically sound, and do the data support the conclusions?

Reviewer #1: Partly

Reviewer #2: Yes

2. Has the statistical analysis been performed appropriately and rigorously? 

Reviewer #1: Yes

Reviewer #2: I Don't Know

3. Have the authors made all data underlying the findings in their manuscript fully available?

Reviewer #1: Yes

Reviewer #2: Yes

4. Is the manuscript presented in an intelligible fashion and written in standard English?

Reviewer #1: Yes

Reviewer #2: Yes

5. Review Comments to the Author

Reviewer #1: Thank you for the opportunity of reviewing this paper. It pertains to an important area, but I think needs some more work before it is ready for publication.

In the methods section there are listed all the patients that have been excluded form study population.

In my opinion it is a simple list and it is not clear on which criteria some patients has been excluded. If some literature references are available, please cite them in the text.

In the discussion section it is explained how the nomogram could be used to obtain the probability of risk of developing depression in patients with hypertension.

However, depression is a multifactorial disease. While nomogram could give a probability risk of developing depression, it could be interesting to briefly explore (maybe revising literature) how it could be implemented in the everyday clinical practice.

Finally, at the end of the discussion section it is mentioned how the study could be affected by a selection bias regarding the study population, please better specify what the authors mean.

I would suggest a reworking of the discussion sections of this paper & then re-submit.

Reviewer #2: The manuscript is well written. However, I have few concerns - (1) Why females are more depressed than males? This need to be discussed in the manuscript. The mean age of subjects is 54.06 years. Therefore, is it associated with menopause? (2) The mean weekly sleep is 7.11 hrs for depressed individuals. If this number is correct, are they also associated with some other sleep related disorder? (3) Discussion part is not very well written. It has to be elaborated and discuss every finding made by the authors.

6. PLOS authors have the option to publish the peer review history of their article (what does this mean?). If published, this will include your full peer review and any attached files.

Reviewer #1: No

Reviewer #2: No

---

## [Author Response · Author response to Decision Letter 0]

17 Mar 2023

Dear Dr. Gregor Stiglic and dear reviewers,

Re: Manuscript ID: PONE-D-22-34347 and Title: Development and validation of a depression risk prediction nomogram for US Adults with hypertension, based on NHANES 2007-2018

Thank you for your letter and the reviewers’ comments concerning our manuscript entitled “Development and validation of a depression risk prediction nomogram for US Adults with hypertension, based on NHANES 2007-2018”. Those comments are valuable and very helpful. We have read through comments carefully and have made corrections. Based on the instructions provided in your letter, we uploaded the file of the revised manuscript. We have marked in red the parts that need to be revised as mentioned by the reviewers. The remaining parts that we have revised are marked in blue. A point-by-point response to the two nice reviewers is listed below this letter. The reviewers' comments are listed below in italics and the specific questions are numbered. Our responses are given in normal type.

We would love to thank you for allowing us to resubmit a revised copy of the manuscript and we highly appreciate your time and consideration.

Sincerely.

Yan, Zhang

15, Mar, 2023

Replies to the reviewers’ comments:

Answers to REVIEWER #1:

Q1. In the methods section there are listed all the patients that have been excluded form study population. In my opinion it is a simple list and it is not clear on which criteria some patients has been excluded. If some literature references are available, please cite them in the text. 

Response：Thank you very much for double checking, this was an oversight on our part, we have listed the inclusion and exclusion criteria for the study population on page 7, lines 10 and 11, and we have also added a flowchart for the screening of study participants (see Fig1). Thank you for the heads up.

Q2. In the discussion section it is explained how the nomogram could be used to obtain the probability of risk of developing depression in patients with hypertension. However, depression is a multifactorial disease. While nomogram could give a probability risk of developing depression, it could be interesting to briefly explore (maybe revising literature) how it could be implemented in the everyday clinical practice.

Response：Thank you for your suggestion, we have added a discussion of how nomogram is implemented in clinical practice on page 29, lines 1 to 9, and cited the "Guide to presenting clinical prediction models for use in clinical settings clinical settings, BMJ (Clinical research ed.) 365 (2019) l737" to explain how the clinical prediction models of nomogram can be applied in clinical practice.

Q3. Finally, at the end of the discussion section it is mentioned how the study could be affected by a selection bias regarding the study population, please better specify what the authors mean.

I would suggest a reworking of the discussion sections of this paper & then re-submit.

Response：Thank you for underlining this deficiency. NHANES is a large database with many indicators, and based on our current limited understanding, we were unable to include all covariates related to hypertension and depression; Therefore, some variables may have been omitted in our selection of included variables and thus would have been subject to selection bias in the selection of variables. We have also included an explanation of selection bias on page 29, lines 13, and page 30, lines 1 to 4.

Answers to REVIEWER #2:

Q1. Why females are more depressed than males? This need to be discussed in the manuscript. 

Response：Thank you for your suggestion. We have further refined the discussion on the relationship between gender and depression on page 25, lines 9 to 13 and page 26, line 1 to 2, and we have also cited "Sex differences in anxiety and depression: role of testosterone, Frontiers in neuroendocrinology 35(1) (2014) 42-57" to explain the mechanism, which mentions that the gender difference in depression between women and men is caused by the different amount of testosterone produced by men and women.

Q2. The mean age of subjects is 54.06 years. Therefore, is it associated with menopause?

Thanks for your great suggestion on improving the accessibility of our manuscript. In our study, depression was mostly concentrated in middle-aged and older adults, and we also found some literature mentioning that older patients are more likely to be depressed. Moreover, the literature “The association of widowhood and living alone with depression among older adults in India, Scientific reports 11(1) (2021) 21641” and “Loneliness and Depression Among Older Adults in Urban Subsidized Housing, Journal of aging and health 30(3) (2018) 458-474” also mention that as people age, the cumulative effects of several adverse factors may increase the risk of negative mental health outcomes, which can eventually cause depression. Moreover, previous studies have concluded that women are more likely to be depressed than men, so women may be more likely to suffer from depression during menopause. We have added discussion on page 25, lines 9 through 13, and page 26, lines 1 through 11, combining the effects of age and gender on depression.

Q3. The mean weekly sleep is 7.11 hours for depressed individuals. If this number is correct, are they also associated with some other sleep related disorder?

Response：We apologize for our carelessness, it was a spelling mistake on my part, I spelled "sleep time on workdays" as "weekly sleep", After a thorough review of the full text, we have corrected all "weekly sleep" to "sleep time on workdays" in the resubmitted manuscript. Thank you for the correction.

Q4. Discussion part is not very well written. It has to be elaborated and discuss every finding made by the authors.

Response：We sincerely thank you for your valuable comments. In the discussion section, we present a more thorough and extensive description of all the nomogram predictors based on our findings. We've discussed and found pertinent literature to discuss for each of the study's findings.

---

## [Decision Letter · Decision Letter 1]

23 Mar 2023

Development and validation of a depression risk prediction nomogram for US Adults with hypertension, based on NHANES 2007-2018

PONE-D-22-34347R1

Dear Dr. Zhang,

We’re pleased to inform you that your manuscript has been judged scientifically suitable for publication and will be formally accepted for publication once it meets all outstanding technical requirements.

Kind regards,

Gregor Stiglic, Ph.D.

Academic Editor

PLOS ONE

Additional Editor Comments (optional):

Reviewers' comments:

Reviewer's Responses to Questions

**Comments to the Author**

1. If the authors have adequately addressed your comments raised in a previous round of review and you feel that this manuscript is now acceptable for publication, you may indicate that here to bypass the “Comments to the Author” section, enter your conflict of interest statement in the “Confidential to Editor” section, and submit your "Accept" recommendation.

Reviewer #1: All comments have been addressed

2. Is the manuscript technically sound, and do the data support the conclusions?

Reviewer #1: Yes

3. Has the statistical analysis been performed appropriately and rigorously? 

Reviewer #1: Yes

4. Have the authors made all data underlying the findings in their manuscript fully available?

Reviewer #1: Yes

5. Is the manuscript presented in an intelligible fashion and written in standard English?

Reviewer #1: Yes

6. Review Comments to the Author

Reviewer #1: Dear Editor,

I have had the opportunity to review the revised manuscript and I am pleased to see that the authors have taken my feedback into account and have made the necessary changes to address the mistakes in the previous version. Overall, I now find their work to be well-written, well-structured, and to present novel and interesting findings that contribute significantly to the field.

The flowchart that has been added to the manuscript clearly explains the inclusion and exclusion criteria for the study population and it results clear and immediate to be understood by readers.

Moreover, the explanation on how the nomogram can be used in clinical practice with its specific literature references add an important scientific value to the text.

The authors have made significant improvements to the methods and discussion sections which were a major concern in my previous review. Specifically, they have provided a detailed explanation of the methods used, which is clear and well-written. The results are now more accurately reported, and the conclusions drawn from them are more robust.

I also appreciate the authors' efforts to address other minor issues that were identified in the previous review.

Overall, I believe that the revised manuscript is a much stronger and sounder piece of research. The authors have addressed as much as possible (especially regarding the selection bias mentioned in the previous review) the statistical mistakes and have made other improvements that have strengthened the manuscript.

7. PLOS authors have the option to publish the peer review history of their article (what does this mean?). If published, this will include your full peer review and any attached files.

Reviewer #1: No

<quillbot-extension-portal></quillbot-extension-portal>

---

## [Editor Report · Acceptance letter]

28 Mar 2023

PONE-D-22-34347R1 

Development and validation of a depression risk prediction nomogram for US Adults with hypertension, based on NHANES 2007-2018 

Dear Dr. Zhang:

I'm pleased to inform you that your manuscript has been deemed suitable for publication in PLOS ONE. Congratulations! Your manuscript is now with our production department. 

Kind regards, 

on behalf of

Dr. Gregor Stiglic 

Academic Editor

PLOS ONE